# TCR Clonality and Genomic Instability Signatures as Prognostic Biomarkers in High Grade Serous Ovarian Cancer

**DOI:** 10.3390/cancers13174394

**Published:** 2021-08-31

**Authors:** Julie Lecuelle, Romain Boidot, Hugo Mananet, Valentin Derangère, Juliette Albuisson, Vincent Goussot, Laurent Arnould, Zoé Tharin, Isabelle Ray Coquard, François Ghiringhelli, Caroline Truntzer, Jean-David Fumet

**Affiliations:** 1Platform of Transfer in Biological Oncology, Georges François Leclerc Cancer Center—UNICANCER, 1 rue du Professeur Marion, 21000 Dijon, France; jlecuelle@cgfl.fr (J.L.); hmananet@cgfl.fr (H.M.); vderangere@cgfl.fr (V.D.); jalbuisson@cgfl.fr (J.A.); fghiringhelli@cgfl.fr (F.G.); ctruntzer@cgfl.fr (C.T.); 2Unité Mixte de Recherche (UMR) INSERM 1231, 7 Boulevard Jeanne d’Arc, 21000 Dijon, France; 3Institut de Chimie Moléculaire Université de Bourgogne (ICMUB) UMR CNRS 6302, 21000 Dijon, France; rboidot@cgfl.fr; 4Department of Biology and Pathology of Tumors, Georges François Leclerc Cancer Center—UNICANCER, 1 rue du Professeur Marion, 21000 Dijon, France; vgoussot@cgfl.fr (V.G.); larnould@cgfl.fr (L.A.); 5Department of Medical Oncology, Georges François Leclerc Cancer Center—UNICANCER, 1 rue du Professeur Marion, 21000 Dijon, France; ztharin@cgfl.fr; 6Laboratoire RESHAPE University Claude Bernard Lyon I, Department of Medical Oncology, Léon-Bérard Center, 28 rue Laennec, 69008 Lyon, France; isabelle.ray-coquard@lyon.unicancer.fr; 7Maison de l’université Esplanade Erasme, University of Burgundy-Franche Comté, 21000 Dijon, France; 8Genomic and Immunotherapy Medical Institute, Dijon University Hospital, 14 rue Paul Gaffarel, 21000 Dijon, France

**Keywords:** HGSC, TCR clonality, HRD, prognostic, biomarkers

## Abstract

**Simple Summary:**

High-grade serous ovarian carcinoma (HGSC) could be analyzed with a molecular stratification defined by different genomic instability signatures associated with specific mutational process and prognostic biomarkers. Immune infiltrate is known to be a robust biomarker in HGSC. We aimed to investigate immune parameters according to genomic instability signatures. We observed that homologous recombination deficiency positive, copy cumber variant signature 7 and TCR (T cells receptor) clonality are good prognostic biomarkers in HGSC. Combining TCR clonality and genomic instability signature or T cell infiltration improved the prognostic value compared to each variable taken alone. We provided a description of immune parameters according to different genomic instability signatures. We identified TCR clonality, alone or combined with genomic instability, as a promising prognostic biomarker in HGSC.

**Abstract:**

Purpose: Immune infiltration is a prognostic factor in high-grade serous ovarian carcinoma (HGSC) but immunotherapy efficacy is disappointing. Genomic instability is now used to guide the therapeutic value of PARP inhibitors. We aimed to investigate exome-derived parameters to assess the tumor microenvironment according to genomic instability profile. Methods: We used the HGSC TCGA (the cancer genome atlas) dataset with genomic characteristics, including homologous recombination deficiency (HRD), copy number variant (CNV) signatures, TCR (T cell receptor) clonality and abundance of tissue-infiltrating immune and stromal cell populations. We then investigated the relationship with survival data. Results: In 578 HGSC patients, HRD status, CNV signature 7 and TCR clonality were associated with longer survival. The combination of high CNV signature 7 expression and HRD status or high CNV signature 3 expression and high TCR clonality was associated with a trend towards longer survival compared to each variable alone. Combining T cell infiltrate and TCR clonality improved the prognostic value compared to T cells infiltration alone. Prognostic value of TCR clonality was confirmed in an independent cohort. Conclusions: TCR clonality is an emerging prognostic biomarker that improves T cell infiltrate information. Analysis of TCR clonality combined with genomic instability could be an interesting prognostic biomarker.

## 1. Introduction

Epithelial ovarian cancer represents the fifth leading cause of death from cancer among women. High-grade serous ovarian cancer (HGSC) is the most aggressive subtype, and accounts for the majority of advanced-stage cases [1]. Despite the gold standard of debulking surgery and platinum/taxane-based chemotherapy, which results in initial response in approximately 75% of patients, most women with HGSC experience relapse and succumb to chemo-resistant disease [2]. Therefore, it is important to understand the mechanisms associated with resistance and to develop novel therapeutic strategies. Currently, two major therapeutics have been developed. Firstly, target therapy including poly-ADP ribose inhibitors (PARPi) and anti-angiogenic treatment (bevacizumab). Around half of HGSC have defects within the homologous recombination DNA repair pathway, called homologous recombination repair deficiency (HRD), and therefore exhibit a distinct clinical profile with a better response to platinum based chemotherapies and greater sensitivity to PARPi [3]. Since 2020, maintenance therapy comprising an association of PARPi and bevacizumab is the recommended strategy for patients with a HRD profile [4]. Thereafter, immunotherapy using immune checkpoint inhibitors (ICI) targeting PD(L)-1 has been evaluated in ovarian cancer. Immunotherapy has transformed cancer treatment in certain solid malignant tumors, such as melanoma [5,6,7], lung cancer [8] and renal cell carcinoma [9]. Epithelial ovarian cancer is controlled by immune response, and high T cell infiltration is associated with better prognosis [10,11,12,13]. However, despite the intrinsic prognostic role of immune infiltrates, most studies testing immunotherapy in HGSC patients failed to show a significant benefit, although a small subset of patients seem to present response and prolonged survival [14,15,16]. There are no robust predictive biomarkers to identify patients who would benefit from immunotherapy as monotherapy [17,18]. Thus, intense research activity is ongoing worldwide to identify predictive biomarkers of efficacy.

Currently, the relationship between HRD as a feature of genomic instability and response to immunotherapy is unclear. It has been reported that HRD cancers exhibit increased immune cell infiltrate compared to HR-proficient (HRP) cancers, especially BRCA mutated tumors [19,20]. However, in the KEYNOTE 100 trial, with pembrolizumab monotherapy for recurrent ovarian cancer, HRD and BRCA status were not found to be associated with response [17]. In a sub-study analysis of the TOPACIO trial, which tested niraparib and pembrolizumab in platinum-resistant HGSC, HRD was not found to be a predictive biomarker [21]. It has been noted that mutational signature 3, previously described by Alexandrov et al. [22] as being related to HRD status, showed an interesting predictive value. 

In 2018, Macintyre et al. [23] provided a stratification of HGSC in seven copy number signatures, based on six fundamental copy number features describing particular genomic instabilities. These signatures are associated with specific known driver gene mutations and signaling pathway modifications. These signatures are also associated with prognosis and predictive response to platinum-based chemotherapy.

To the best of our knowledge, there is a lack of data about immune infiltrate based on genomic instability. The aim of this study was to use exome-derived parameters to assess the tumor microenvironment according to genomic instability profile.

## 2. Materials and Methods

### 2.1. Study Population 

#### 2.1.1. TCGA Data 

Five hundred and seventy-eight patients with high-grade serous ovarian cancer and complete survival data were extracted from The Cancer Genome Atlas (TCGA) (https://www.cancer.gov/tcga, accessed date: v27.0, 29 October 2020). Patients were described with different variables taken from different publications, namely the number of TCR clones [24] and the HRD score [25] (Appendix A). CNV signatures were inferred following the methodology of Macintyre et al. [23]. Briefly, CNV signature 1 was described to be associated with poor prognosis related to oncogenic RAS-MAK pathway. CNV signature 2 was also associated with poor prognosis and related to tandem duplication through CDK12 inactivation. CNV signature 3 was associated with good overall survival and related to BRCA 1-2 related HRD. CNV signature 4 was related to whole genome duplication due to failure of cell cycle. CNV signature 5 was correlated with a number of chromothriptic-like events. CNV signature 6 was associated with focal amplification due to failure of cell cycle control. CNV signature 7 was associated with non BRCA 1-2 related HRD. The abundance of 10 tissue-infiltrating immune and stromal cell populations (CD3+ T cells, CD8+ T cells, cytotoxic lymphocytes, NK (Natural Killer) cells, B lymphocytes, monocytic lineage, myeloid dendritic cells, neutrophils, endothelial cells and fibroblasts) were estimated using transcriptomic profiles obtained through the R library TCGA2STAT [26] (Agilent G450 microarray expression) and the Microenvironment Cell Populations-counter (MCP-counter) method [27]. Somatic mutations were extracted using the R package MADGiC [28].

#### 2.1.2. CGFL Data

In this single-center, retrospective study, we included 31 patients with HGSC for whom whole exome sequencing (WES) analysis was performed as part of routine care, and interpreted by the Molecular Tumor Board of the Georges François Leclerc Cancer Center (CGFL). Patients were followed between January 1997 and August 2020. WES analyses were performed during first-line therapy. WES analysis is performed in routine care in our center in order to identify potentially targetable mutations for second-line therapy. Before patients consented to WES of their tumoral tissue, they were informed by their oncologist. Germline testing was performed after counseling by a clinical geneticist.

Only patients for whom written informed consent was obtained and recorded in the medical chart were included in this study. The study was approved by the CNIL (French national commission for data privacy) and the local ethics committee, and was performed in accordance with the Helsinki Declaration and European legislation.

### 2.2. Methodology for the CGFL Data 

#### 2.2.1. Sample Selection

Archival tumor sample (primary or metastasis) or a new tumor biopsy (at the physician’s discretion) was selected by clinicians for genomic analysis. Tumor cellularity was assessed by a senior pathologist on a hematoxylin and eosin slide from the same biopsy core used for nucleic acid extraction and molecular analysis.

#### 2.2.2. DNA Isolation

DNA was isolated from archival tumor tissue using the Maxwell 16 FFPE Plus LEV DNA purification kit (Promega, Madison, WI, USA). DNA from whole blood (germline DNA) was isolated using the Maxwell 16 Blood DNA Purification kit (Promega) following the manufacturer’s instructions. The quantity of extracted genomic DNA was assessed by a fluorometric method with a Qubit device.

#### 2.2.3. Whole Exome Capture and Sequencing

Two hundred ng of genomic DNA were used for library preparation, using the Agilent SureSelectXT reagent kit (Catalog number G9642B, Agilent Technologies, Inc., Santa Clara, CA, USA) and the All Exon v5 probeset (5190–8863, Agilent Technologies, Inc.). Following hybridization, the libraries were purified according to the manufacturer’s recommendations and amplified by polymerase chain reaction (12 cycles). DNA integrity was verified using TapeStation (SCREENTAPE D1000 tapestation 5067–5582, reagents D1000 tapestation 5067–5583). Concentrations were measured using Qubit^®^ dsDNA BR Assay Q32853 (Thermo Fisher Scientific, Waltham, MA, USA). Loading concentrations were 22 nM for fragmentation and 6 pM for NextSeq.

#### 2.2.4. Exome Analysis Pipeline

Reads in FASTQ format were aligned to the reference human genome GRCh37 using the Burrows–Wheeler aligner (BWA v.0.7.17, http://bio-bwa.sourceforge.net/, accessed on 29 August 2021). Local realignment and duplicate reads were performed using the Genome Analysis Toolkit (GATK v.4.1.3.0, https://gatk.broadinstitute.org/hc/en-us, accessed on 29 August 2021). Somatic single nucleotide variants (SNVs) were called with VarScan (v2.4.3 http://varscan.sourceforge.net/, accessed on 29 August 2021) [29] and Mutect (v1.1.7, https://software.broadinstitute.org/cancer/cga/mutect, accessed on 29 August 2021) [30], insertion/deletions (indels) were called with VarScan and Strelka (v2.9.2, https://github.com/Illumina/strelka, accessed on 29 August 2021) [31]. TITAN [32] was used to infer copy number alterations (CNA). TCR clones were inferred using MixCR [33] and HRD score was calculated with ScarHRD [34].

### 2.3. Statistical Analysis 

Using the Chi-2 or Fisher’s exact test for qualitative variables or the Wilcoxon test for continuous variables, clinical and genomic characteristics of patients were compared between the TCGA and CGFL cohorts.

Correlations between continuous variables were quantified using Pearson’s correlation coefficient. *p*-values were adjusted using Benjamini–Hochberg FDR (false discovery rate) correction [35]. All adjusted *p*-values less than 0.1 were considered statistically significant.

Survival analysis was performed using the survival R library. Continuous variables were dichotomized using the methodology of Lausen et al. via the maxstat library [36]. The prognostic value of the different variables was tested using univariate Cox regression models for overall survival (OS). Age was tested as a confounding factor by testing interactions between each variable of interest and age. Survivors were censored at 5 years. Survival probabilities were estimated using the Kaplan–Meier method and survival curves were compared using the log-rank test. 

Statistical analyses were performed using the R software (http://www.R-project.org/, version 4.0.3, accessed on 29 August 2021) and graphs were drawn using GraphPad Prism version 9.02.

## 3. Results

### 3.1. Patient Characteristics

Five hundred and seventy-eight patients with HGSC taken from the TCGA ovarian dataset were included in this analysis. The median follow-up was 45.5 months (Interquartile Range (IQR) = 49.3 months). The median age at diagnosis was 59 years (IQR = 18). The population was mostly composed of stage III (78.5%) and stage IV (14.5%). The main characteristics of the population are reported in Table 1. The patients displayed characteristics typical of the HGSC population. 

### 3.2. Prognostic Value of Genomic Instability Signatures

#### 3.2.1. HRD Profile

We observed that 55.6% of 171 HGSC, for whom HRD status was available, exhibited a HRD profile using a classic cutoff of 42 (Figure 1A). As expected, these patients had longer survival than HRP patients (median OS, respectively, 49.8 months (IQR = 54.6) and 38.1 months (IQR = 31.9)); HR = 0.61 [0.39; 0.95], *p* = 0.03) (Figure 1B).

#### 3.2.2. CNV Signatures

Among 413 HGSC from TCGA evaluated for CNV signatures, we observed that CNV signature 7 (known to be related to non-BRCA 1-2 HRD scar) was associated with improved survival (HR = 0.21 [0.06; 0.73], adjusted *p*-value = 0.1). CNV signature 3 seemed to be associated with a non-statistical improved survival (HR = 0.51 [0.24; 1.04], adjusted *p*-value = 0.16), while CNV signature 6 showed a trend towards poorer OS (HR = 4.59 [0.9; 23.4], adjusted *p*-value = 0.16). In this TCGA cohort, other CNV signatures did not impact survival (Table 2).

#### 3.2.3. Combined Analysis of HRD Status and CNV Signatures

When examining the relation between CNV signatures and HRD profile, we found that HRD tumors exhibited a higher proportion of CNV signatures 3 and 7 (adjusted *p*-value < 1 × 10^−3^), whereas HRP tumors exhibited a higher proportion of CNV signatures 2 (adjusted *p*-value = 0.02) and 6 (adjusted *p*-value = 0.09) (Figure 1C). Among the 11 patients with a BRCA mutation, only one patient did not have a HRD profile. This patient had expression of CNV3 and not CNV7, while the majority of the other BRCA mutated patients had a HRD, CNV3 plus CNV7 profile (Figure 1A). Combining dichotomized CNV signature 7 level (cutoff equal to 0.14) and HRD status was non-redundant, and we observed a trend towards better survival in HRD/CNV7^High^ patients than in other patients (HR = 0.54 [0.28; 1.02], *p* = 0.06) (Figure 1D). Furthermore, the benefit of CNV signature 7 information seemed to be limited to HRD tumors. Median survival was 39.6 (IQR = 48.6) months in CNV7^Low^ status versus 71.7 (IQR not reached) months in CNV7^High^ status (HR = 0.61 [0.29; 1.31], *p* = 0.21), while in HRP tumors, median survival was equivalent, at 38 (IQR = 40.5) and 37.3 (IQR = 37.6) months (HR = 0.8 [0.36; 1.82], *p* = 0.6). It has been noted that combination of HRD status and CNV 3 signature did not improve survival compared with HRD or CNV 3 signature alone.

### 3.3. Prognostic Value and Relationship between Genomic Instability Signatures and Immune Population

Among 566 assessable HGSOC, we evaluated the prognostic value of tissue-infiltrating immune and stromal cell populations. Using univariate Cox models, only the presence of neutrophils and fibroblasts was statistically associated with worse survival (respectively, HR = 1.87 [1.32; 2.63], adjusted *p*-value = 0.09 and HR = 1.14 [1.02; 1.28], adjusted *p*-value = 0.001) (Table 3). We showed that fibroblasts had a deleterious effect in HRP tumors (HR = 2.32 [1.23; 4.4], *p* = 0.01) (Figure 2A). Likewise, HRD/Neutrophils^Low^ exhibited a better survival that HRP/Neutrophils^High^ (HR = 0.45 [0.25; 0.81], *p* = 0.008) (Appendix A).

We then investigated whether immune population abundances were different according to HRD status. We showed that HRD status was significantly associated with a higher abundance of endothelial cells (adjusted *p*-value = 0.08) (Figure 2B). 

Subsequently, we analyzed the association between pathogenic BRCA 1-2 mutations (Appendix A) and the expression of immune cell populations. We did not observe a significant adjusted difference in BRCA 1-2 mutant tumors in comparison to wild type tumors (Figure 2C). High infiltration of each immune cell population was not associated with overall survival in the BRCA 1-2 mutant group or the wild type group.

To investigate further, we assessed the association between immune population abundances and CNV signature levels (high vs. low). We observed that patients with a high proportion of CNV signatures 1 or 6 respectively had a higher abundance of myeloid dendritic cells (adjusted *p*-value = 0.08) and B lineage cells (adjusted *p*-value = 0.9). Patients with a low proportion of CNV signatures 2 or 6 had a higher abundance of myeloid dendritic cells (respectively, adjusted *p*-value = 0.02, 0.05). Patients with a low proportion of CNV signature 6 had a higher abundance of NK cells (adjusted *p*-value = 0.05) (Figure 2D). Survival analysis added no relevant information. 

Moreover, we analyzed the relation between CNVs of 43 available oncogenes or tumor suppressor genes (amplification vs. no amplification and deletion vs. no deletion) and each of the CNV signatures expressions (Low vs. High) (Appendix A). 

### 3.4. TCR Clonality 

Beyond T cell infiltrate, we investigated whether TCR clonality could have a prognostic value. TCR clonality reflects the presence of a high number of clonal T cells, thus, suggesting the presence of tumor-specific T cells. TCR clonality was significantly correlated with the immune cell population (Appendix A). We observed a median TCR clonality of 3 (IQR = 5). Using a best cutoff of 11, we showed that patients with high number of TCR clones had significantly better survival (median OS of 64.9 months (IQR = 34.1) for TCR^High^ patients vs. 44.81 (IQR = 50.1) months for TCR^Low^ patients; HR = 0.45 [0.23; 0.89], *p* = 0.02) (Figure 3A). 

When we focused our analysis on highly T cell infiltrated tumors, we observed that TCR clonality remained prognostic (median OS in TCR^High^/T cells^High^ tumors was 58 months (IQR not reached), and in TCR^Low^/T cells^High^ tumors 43.4 months (IQR = 45.5), HR = 0.5 [0.24; 1.01], *p* = 0.05) (Figure 3B). This suggests the importance of the TCR clonality event in patients richly invaded by immune cells. 

To investigate further, we combined HRD and TCR clonality. We observed that patients with a higher number of TCR clones tended to have better survival than TCR^Low^, independent of HRD status (Figure 3C).

Finally, we combined BRCA status and TCR clonality and observed a subgroup, BRCA 1-2 WT tumors with a low number of TCR clones, who had a significantly poorer survival than other patients (HR = 1.86 [1.01; 3.4], *p* = 0.04) (Figure 3D). 

Regarding CNV signatures, no significant adjusted difference was observed (Figure 3E). When we combined TCR clonality and CNV signature levels, we found that patients with a high proportion of CNV signature 3 and a high number of TCR clones had better survival than other patients (*p* = 0.02) (Figure 3F). Similarly, we found that patients with CNV 7^Low^ signature and a low number of TCR clones had worse survival (HR = 2.22 [1.41; 3.51], *p* < 1 × 10^−3^) (Figure 3G). 

### 3.5. Same Analysis in an Independent Cohort

We performed our methodology in an independent HGSC cohort of 31 patients, treated in a single center, enriched in stage IV FIGO and including older patients than in the TCGA cohort (Table 1). Similarly, we used exome sequencing to link genomic instability and TCR clonality to prognosis.

We observed that 70% of this population were HRP (Figure 4A). HRD profile was not associated with survival in this cohort (*p* = 0.39) (Figure 4B). Concerning CNV signatures, signatures 1 and 5 were expressed more than in the TCGA cohort (adjusted *p*-value < 1 × 10^−3^), whereas CNV signatures 3 and 6 were expressed less than in the TCGA cohort (respectively, adjusted *p*-values < 1 × 10^−3^ and 0.06) (Table 1). CNV signatures were not associated with survival in this cohort (Appendix A). In concordance with previous published data and TCGA, CNV signatures 1 and 7 were respectively, significantly negatively and positively correlated with HRD profile (Figure 4C). Similarly, to TCGA, HRD tumors with a high CNV signature 7 status tended to have better survival than HRD/CNV7^Low^ status (HR = 0.29 [0.07; 1.12], *p* = 0.07) (Appendix A).

When we focused on TCR clonality, we showed that TCR^High^ (defined as more than 10 clones) was significantly associated with better survival (median OS for TCR^High^ tumors was 212.8 (IQR = 79.7) months vs. 45.9 (IQR = 33.8) for TCR^Low^ tumors) (Figure 4D). 

Combining TCR clonality with HRD status did not improve prognostic value in comparison with TCR clonality alone (Figure 4E). Similarly, the combination of TCR clonality score with CNV signatures did not improve the prognostic value of TCR clonality alone (data not shown). However, when combining TCR clonality and BRCA 1-2 mutation status, we observed the same subgroup that, in TCGA cohort (WT/TCR^Low^ tumors), had a significantly poorer survival than other (HR = 8.8 [1.1; 70.3], *p* = 0.04) (Figure 4F).

## 4. Discussion

Genomic instability, especially HRD, is now a major prognostic and predictive parameter for treatment response (platinum-based chemotherapy and PARP inhibitors) in ovarian cancer in first-line setting. We confirm here that HRD positive tumors have a better prognosis than HRP tumors. We also show that CNV signature 7, correlated with the HRD profile, was associated with a good prognosis. Interestingly, these two parameters (HRD and CNV signature 7) were not redundant, and the combination of both was a marker of better prognosis. Future analyses could evaluate the value of this combination in assessing the prediction of response to platinum-based chemotherapy and PARPi. HRP tumors seem to be associated with CNV signatures 2 and 6; additionally, CNV signature 6 is known to be related to the failure of cell cycle control. These data suggest that cell cycle inhibitors could be an interesting strategy in HRP tumors with high CNV signature 6. Adavosertib, a WEE1 inhibitor, is currently being tested in advanced ovarian cancer [37]. It will be interesting to explore the predictive role of CNV signature 6 with this treatment.

Furthermore, it has been reported that genomic instability is responsible for higher immune-monitoring, related to the higher production of neoantigens [38]. Accordingly, several studies have shown that HRD tumors were more infiltrated with immune cells [19,20]. Here, we confirmed these results, in particular in BRCA mutated tumors with significant infiltration of cytotoxic lymphocytes, Natural Killer (NK) cells and fibroblasts. Furthermore, we showed that HRD tumors were enriched in endothelial cells. If we hypothesize that HRD tumors are more vascularized, then these data may strengthen the rationale for combining PARPi and bevacizumab in these tumors as reported in the first-line setting [4]. Of note, in HRP patients, fibroblasts are a marker of poor prognosis. We found here a potential mechanism of immuno-exclusion, which constitutes a basis for the development of immunotherapy in association with fibrosis inhibitors. Combined treatments with ICI + anti TGFb [39] could be a promising option for positioning immunotherapy in these HRP populations. 

When we looked more specifically at CNV signatures, we found that CNV signature 3 was characterized by NK infiltration, while signature 6 was characterized by a low proportion of NK. Furthermore, CNV signature 7 was characterized by a low proportion of neutrophils. To the best of our knowledge, this is the first description of the proportion of immune cells according to CNV signatures. The high presence of NK cells in signature 3, known to be correlated with BRCA mutation status, reveals a new potential target for using immunotherapy in these tumors. Indeed, there are currently treatments under development such as monalizumab [40], an inhibitor of NKG2A, which can lift the inhibition on NKs. These data will need to be confirmed in other cohorts and by other techniques.

In addition to the estimated proportions of immune cells, complex exome analysis enabled the evaluation of TCR clonality. We show that TCR clonality is a prognostic factor associated with prolonged survival. Interestingly, the combined analysis of cytotoxic lymphocyte proportion and TCR clonality improved the prognostic value over the lymphocyte infiltrate alone. It has been shown that about 10% of the CD8 infiltrate in ovarian cancer is tumor-specific [41]. This important information showed that the quality of the infiltrate is as important as the quantity. A recent publication has shown that the prognostic significance of TILs in ovarian cancer was dictated by T cell clonality and that some TIL-low tumors could have high TCR clonality and may be an interesting candidate for ICI [42]. This was a major breakthrough in the context of ovarian cancers, where ICI is disappointing with response rates of about 10–15% [14,15,16]. Only a small group of patients seemed to benefit from ICI for any significant length of time. TCR clonality appears to be a promising new factor for selecting patients who may benefit from ICI immunotherapy. 

Furthermore, according to our combined analysis, a high CNV signature 3 proportion (related to BRCA 1-2 mutations) combined with a high number of TCR clones was the strongest predictor of increased survival. This more specific analysis of T infiltrate, according to their TCR diversity in these HRD populations, is an interesting element for future biomarkers in HGSC. Multiple immune cell populations are known to reduce in size as age increases, while the number of somatic mutations increases with age. This hypothesis was tested in our data, but no interaction was found between age and corresponding variables of interest.

We are aware of some limitations in this study. First, the technical limitations due to the small number of patients, especially in the validation set. Second, the lack of accurate data on chemotherapy sequences may lead to a bias because the impact of chemotherapy on the intra-tumor immune microenvironment cannot be assessed. Despite these limitations, we reported a link between genomic instability and the tumor microenvironment. These hypotheses can be used to define new options for the future, most notably incorporating some ICI in some subgroups (HRD/T cells^High^ /TCR^High^), and define new options for the HRP populations for use in addition to the bevacizumab. 

## 5. Conclusions

We provided a description of the immune landscape in HGSC, based on exome-derived immune parameters, according to genomic instability. TCR clonality is a promising prognostic biomarker which improved the prognostic value of T cells infiltrate and genomic instability signature. 

## Figures and Tables

**Figure 1 cancers-13-04394-f001:**
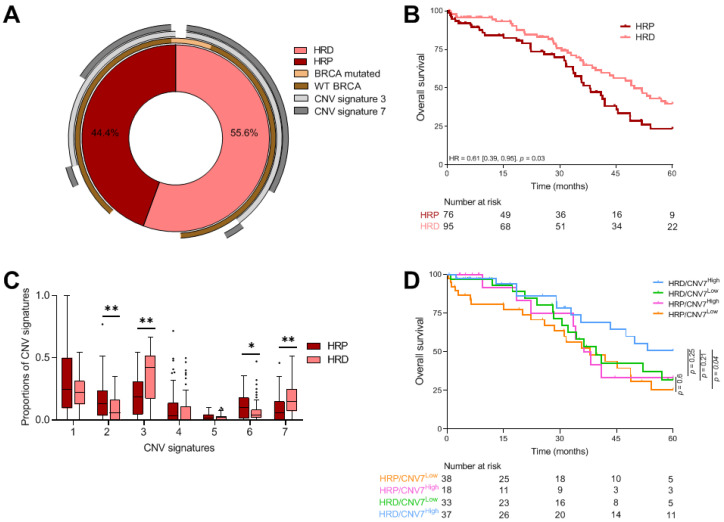
Genomic description of TCGA cohort. (**A**) Donut showing percentage of TCGA HGSC patients that were HRD or HRP combined with BRCA status and CNV signatures 3 and 7 proportions above 0. (**B**) Kaplan–Meier curves with patients stratified according to HRD profile for overall survival. (**C**) Boxplots showing the proportion of CNV signatures for patients stratified according to HRD profile. Adjusted *p*-values < 0.1 are represented by a star, and adjusted *p*-values < 0.05 by a double star. (**D**) Kaplan–Meier curves with patients stratified according to HRD profile and CNV signature 7 proportion (the cutoff is 13.6%). HRD: homologous recombination deficiency; HRP: homologous recombination proficient; CNV: copy number variant. * *p* ≤ 0.05; ** *p* ≤ 0.01.

**Figure 2 cancers-13-04394-f002:**
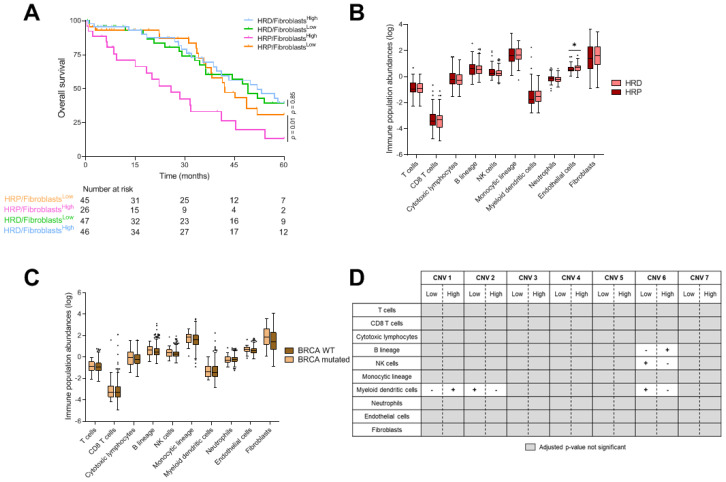
Genomic instability signatures and immune population. (**A**) Kaplan–Meier curves with patients stratified according to the HRD profile and the abundance of fibroblasts for overall survival. (**B**,**C**) Boxplots showing immune population abundances according to the HRD profile (**B**) and the BRCA status (WT vs mutated) (**C**). Adjusted *p*-values < 0.1 are represented by a star. (**D**) Table summarizing comparisons of population abundances according to CNV signatures. Adjusted *p*-values < 0.1 were considered significant. The “+” (respectively, “−“) symbol means an enrichment (respectively, decrease) of given population in the corresponding category. HRD: homologous recombination deficiency; WT: wild-type; CNV: copy number variant * *p* ≤ 0.05.

**Figure 3 cancers-13-04394-f003:**
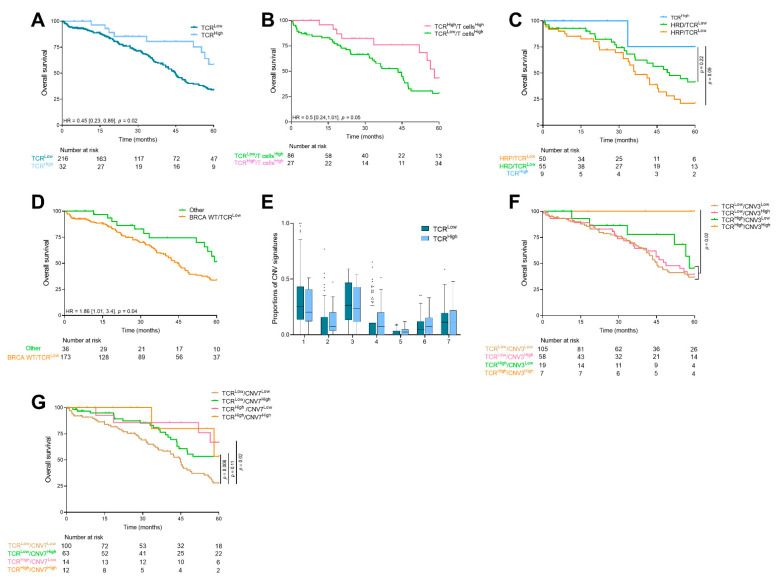
Description of TCR clonality. Kaplan–Meier curves for overall survival with patients stratified according to (**A**) TCR clonality, (**B**) TCR clonality and the abundance of T cells, (**C**) TCR clonality and HRD profile and (**D**) TCR clonality and BRCA status. The cutoff for defining High or Low TCR clonality was 11. (**E**) Boxplots showing the proportion of CNV signatures according to the TCR clonality (cutoff = 11). Kaplan–Meier curves for overall survival with patients stratified according to TCR clonality and (**F**) CNV signature 3 proportion, and (**G**) CNV signature 7 proportion. TCR: T cells receptor; HRD: homologous recombination deficiency; CNV: copy number variant.

**Figure 4 cancers-13-04394-f004:**
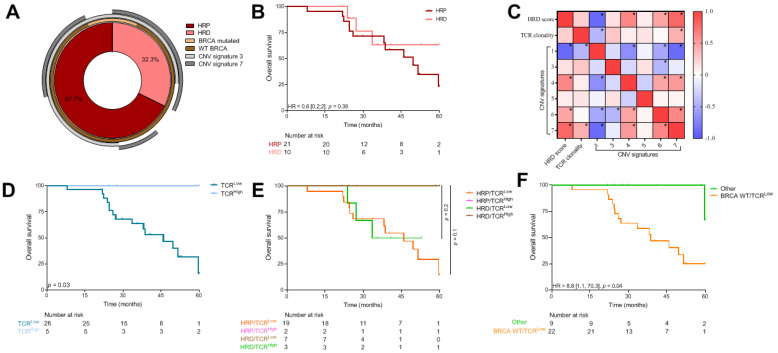
TCR clonality and CNV signatures. (**A**) Percentage of patients from the CGFL cohort that were HRD or HRP combined with BRCA status and CNV signatures 3 and 7 proportions above 0. (**B**) Kaplan–Meier curves with patients stratified according to HRD profile for overall survival. (**C**) Heatmap showing the correlation matrix between HRD score, TCR clonality and CNV signatures; correlations were calculated with Pearson’s correlation coefficient and *p*-values < 0.05 are represented by a star. Kaplan–Meier curves for overall survival with patients stratified according to (**D**) TCR clonality, (**E**) HRD profile and TCR clonality and (**F**) TCR clonality and BRCA status. HRD: homologous recombination deficiency; HRP: homologous recombination proficient; CNV: copy number variant; TMB: tumor mutational burden.

**Table 1 cancers-13-04394-t001:** Summary of clinical and genomic characteristics of patients from the TCGA and CGFL cohorts.

Variables		CGFL Cohort (*n* = 31)	TCGA Cohort (*n* = 578)	*p*-Value	Adjusted * *p*-Value
Clinical characteristics	Stage, *n* (%)			0.005	0.01
	III	20 (64.5)	384 (78.5)		
	IV	11 (35.5)	72 (14.7)		
	Other	0 (0)	33 (6.7)		
	Age at diagnosis, median (IQR)	64.8 (13.73)	59 (18)	0.06	0.06
	Age at diagnosis, *n* (%)				0.06
	≤60 years	11 (35.5)	267 (54)	0.06	
	>60 years	20 (64.5)	225 (46)		
	Overall survival (months), median (IQR)	49.6 (145.8)	45.5 (49.3)	-	
Genomic characteristics	HRD score, median (IQR)	34 (35)	44 (31)	0.04	0.06
	HRD score, *n* (%)				
	<42	10 (32.3)	76 (44.4)	0.03	0.06
	≥42	21 (67.7)	95 (55.6)		
	TCR clones, median (IQR)	6 (4)	3 (5)	0.002	0.01
	CNV signature 1, median (IQR)	0.57 (0.51)	0.25 (0.28)	<1 × 10^−3^	<1 × 10^−3^
	CNV signature 2, median (IQR)	-	0.07 (0.17)	-	-
	CNV signature 3, median (IQR)	0.11 (0.17)	0.26 (0.34)	<1 × 10^−3^	<1 × 10^−3^
	CNV signature 4, median (IQR)	0.02 (0.15)	0 (0.12)	0.56	0.56
	CNV signature 5, median (IQR)	0.05 (0.02)	0.01 (0.03)	<1 × 10^−3^	<1 × 10^−3^
	CNV signature 6, median (IQR)	0.03 (0.06)	0.04 (0.11)	0.04	0.06
	CNV signature 7, median (IQR)	0.05 (0.25)	0.11 (0.18)	0.38	0.45

Continuous variables were described by median values and interquartile range (IQR). Categorical variables were described by number of observation and percentages (%). CNV signature 2 was not expressed in the CGFL cohort. * *p*-values were adjusted using Benjamini–Hochberg FDR correction. HRD: homologous recombination deficiency; TCR: T cell receptor; CNV: copy number variant.

**Table 2 cancers-13-04394-t002:** Univariate Cox models for overall survival in the TCGA cohort and CNV signatures.

Variables	HR	95%CI	*p*-Value	Adjusted * *p*-Value
CNV signature 1	1.3	[0.71; 2.4]	0.4	0.47
CNV signature 2	2.11	[0.63; 6.99]	0.22	0.31
CNV signature 3	0.51	[0.24; 1.04]	0.06	0.16
CNV signature 4	1.94	[0.77; 4.9]	0.16	0.28
CNV signature 5	1.48	[0.01; 174.54]	0.87	0.87
CNV signature 6	4.59	[0.9; 23.4]	0.07	0.16
CNV signature 7	0.21	[0.06; 0.73]	0.01	0.10

* *p*-values were adjusted using Benjamini–Hochberg FDR correction. HR: hazard ratio; CI: confidence interval; CNV: copy number variant; FDR: false discovery rate.

**Table 3 cancers-13-04394-t003:** Univariate Cox model for overall survival in the TCGA cohort and immune cell populations.

TILS	HR	95%CI	*p*-Value	Adjusted * *p*-Value
T Cells	0.98	[0.79; 1.21]	0.85	0.95
Cd8 T Cells	1.02	[0.89; 1.16]	0.82	0.95
Cytotoxic Lymphocytes	1.05	[0.88; 1.25]	0.61	0.95
B Lineage	0.89	[0.72; 1.1]	0.28	0.67
Nk Cells	1.02	[0.76; 1.37]	0.89	0.95
Monocytic Lineage	1.08	[0.92; 1.27]	0.34	0.67
Myeloid Dendritic Cells	0.99	[0.85; 1.17]	0.95	0.95
Neutrophils	1.87	[1.32; 2.63]	<1 × 10^−3^	0.001
Endothelial Cells	1.40	[0.97; 2.02]	0.07	0.24
Fibroblasts	1.14	[1.02; 1.28]	0.02	0.09

* *p*-values were adjusted using Benjamini–Hochberg FDR correction. HR: hazard ratio; CI: confidence interval; NK: natural killer; FDR: false discovery rate.

## Data Availability

CGFL Data are available from authors upon reasonable request.

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
