# Peer review of "TCR Clonality and Genomic Instability Signatures as Prognostic Biomarkers in High Grade Serous Ovarian Cancer"

_cancers, 2021, doi:10.3390/cancers13174394_

Round 1

Reviewer 1 Report

The manuscript titled “TCR clonality and genomic instability signatures as prognostic biomarkers in high grade serous ovarian cancer” investigated the potential of several molecular features as biomarkers for high grade serous ovarian carcinomas (HGSC). Using TCGA datasets and WES data from a small cohort of 31 patients from a single centre, the authors analysed the association between survival and TCR clonality, HRD scores, CNV signatures, immune cell population and somatic mutations. The parameters investigated are relevant to HGSC. The study concluded that TCR clonality is a good prognostic marker for HGSC survival. Combination of genomic instability and TCR clonality further improved the prognostic power. The manuscript was well-written, but not without some amendments to consider.

  1. When describing the TCGA dataset included in the analysis, please describe the treatment status of these patients. Chemotherapy is known to affect the size of immune cell populations. It is thus important to take into account when investigating the potential of this parameter as a biomarker.
  2. Likewise, age is also a confounding factor in this type of analysis. Multiple immune cell populations are known to reduce in size as age increases, while number of somatic mutations increases with age. Please demonstrate ways to correct for these factors by stratification, or multivariate analysis.
  3. CNV signatures 3 and 7 were described but it may be worth briefly describing the other 5 CNV signatures as well.
  4. In terms of the analyses involving multiple variables e.g. Tables 2 and 3, the adjusted p value should be considered in the interpretation of results. The issue lies in multiple testing where type I error is inflated. It is noted that correction was made using the Benjamini-Hochberg method and really it is this adjusted p value that should be used to conclude statistical significance.

Reviewer 2 Report

Lecuelle J. et al. study is focused on demonstrating the translational significance of the homologous recombination deficiency (HRD) status vs the proficient (HRP) status, of the high TCR clonality and the CNV signatures 3 or 7, as promising biomarkers of good prognosis in HGSC. While the authors analyzed a large cohort of HGSC patients from TGCA database, they performed the whole exome sequencing (WES) only in 31 patients. Although the study displays innovative potential and the experimental approach is strongly supported by valid data, this reviewer believes that the following concerns need to be addressed:

  • Although the stratification of the 7 CNV signatures has been reported in Macintyre et al. study, in the Abstract, Results and Discussion sections the authors should better describe, explain and comment on the significance of the CNV signatures and in particular of the number 3 and 7 in accordance with their obtained results. The authors should discuss the differences between all the CNV signatures and whether CNVs of genes already known to be associated with tumorigenesis in these tumors are correlated or in accordance with their findings.
  • It may be useful to add a scheme or graphical model with the summary of these data, particularly focused on the different immune cell populations enriched in the CNV signatures and the potential associated CNVs in oncogenes or tumor suppressor genes.
  • In the figure legend of Table/Figure 2D, the authors should explain + and - symbols in the white boxes.

Reviewer 3 Report

The work presented by Lacuelle et al, titled “TCR clonality and genomic instability signatures as prognostic biomarkers in high grade serous ovarian cancer” is overall interesting and important in its matter.  However, the work seems still preliminary and should be improved before acceptance.

See my comment below:

The authors report in their comparisons the adjusted P-value, however most of the time they do not mention it. Also, they focus their attention on comparison not significant by adjusted p-value. This will need at least to be mention.

What cut-off has been used to define the CNV7 signature High and Low?

HRD/CNV3 comparison should be also reported in figure 1. Or at least discussed.

The finding of Fibroblast infiltrate and the correlation with HRP HRD is interesting, however, the authors did not mention Neutrophils that also seems highly significant. Same comparisons especially Kaplan Meier’s should be shown also for them.

The TCR clonality analysis is very interesting, however, the fact that the number of patients is very low makes the data weak, which is reflected in some barely significant P values. Especially in Figure 4.

For this reason, I find the title firmly highlighting TCR clonality as a new biomarker not appropriate and should be rewritten highlighting the still preliminary feature of the data shown in the manuscript.

Authors should also discuss their findings in view of this recent paper (Tsuji T., Eng K. H., Matsuzaki J., Battaglia S., Szender J. Brian, Miliotto A., Gnjatic S., Bshara W., Morrison C. D., Lele S., Emerson R. O., Wang J., Liu S., et al Clonality and antigen-specific responses shape the prognostic effects of tumor-infiltrating T cells in ovarian cancer. Oncotarget. 2020; 11: 2669-2683.) done analyzing as well the TCGA cohort that suggests opposite results.

Round 2

Reviewer 3 Report

I am satisfied with the Author's answers.